# Mathematical Modelling of Ree-Eyring Nanofluid Using Koo-Kleinstreuer and Cattaneo-Christov Models on Chemically Reactive *AA*7072-*AA*7075 Alloys over a Magnetic Dipole Stretching Surface

**Zahir Shah** [1] **, Narcisa Vrinceanu** [2,*] **, Muhammad Rooman** [1]**, Wejdan Deebani** [3] **and Meshal Shutaywi** [3]

[1] Department of Mathematical Sciences, University of Lakki Marwat, Lakki Marwat 28420, Pakistan; zahir@ulm.edu.pk (Z.S.); roomankhan31@gmail.com (M.R.)
[2] Department of Industrial Machines and Equipments, Faculty of Engineering, "Lucian Blaga" University of Sibiu, 10 Victoriei Boulevard, 550024 Sibiu, Romania
[3] Department of Mathematics, College of Science & Arts, King Abdulaziz University, P.O. Box 344, Rabigh 21911, Saudi Arabia; wdeebani@kau.edu.sa (W.D.); mshutaywi@kau.edu.sa (M.S.)
*   Correspondence: vrinceanu.narcisai@ulbsibiu.ro

**Abstract:** In the current study, since nanofluids have a high thermal resistance, and because non-Newtonian (Ree-Eyring) fluid movement on a stretching sheet by means of suspended nanoparticles *AA*7072-*AA*7075 is used, the proposed mathematical model takes into account the influence of magnetic dipoles and the Koo-Kleinstreuer model. The Cattaneo-Christov model is used to calculate heat transfer in a two-dimensional flow of Ree-Eyring nanofluid across a stretching sheet, and viscous dissipation is taken into account. The base liquid water with suspended nanoparticles *AA*7072-*AA*7075 is considered in this study. The PDEs are converted into ODEs by exhausting similarity transformations. The numerical solution of the altered equations is then performed utilising the HAM. To examine the performance of velocity, temperature profiles, concentration profiles, skin friction, the Nusselt number, and the Sherwood number, a graphical analysis is carried out for various parameters. The new model's key conclusions are that the *AA*7075 alloy outperforms the *AA*7072 alloy in terms of thermal performance as the volume fraction and ferro-magnetic interaction constraint rise. Additionally, the rate of heat transmission and the skin friction coefficient improve as the volume fraction rises.

**Keywords:** Ree-Eyring nanofluid; magnetic dipole; viscous dissipation; Cattaneo-Christov model; Koo-Kleinstreuer model; chemical reaction

## 1. Introduction

The progressive thermal patterns of nanoparticles have an extensive range of exploitations in the engineering, industrial, technical, and biomedical fields. Many thermal engineering and industrial processes employ nanofluids to increase their thermal efficiency. In recent decades, dynamic scientists have shown interest in nanoparticles with a small size (1–100 nm). Nanofluids are nanoparticle suspensions in base fluids. It is noted that these particles do not change the reaction process, but they do improve the fundamental thermal processes of base liquids at the peak level. Nanoparticles are used in sophisticated thermal extrusion systems, engineering heating devices, biomedical applications, cancer treatments, the chemotherapy process, energy resources, heat exchangers, manufacturing processes, thermal management equipment, and many other applications. Usually, these nanoparticles undergo aggregation so that a fluid can flow through a porous medium as a completely interconnected network (ideal porous pipe), formed by the constricted channel between each pore. Choi [1] proposed a ground-breaking study on the thermal characteristics of nanofluids, prompting other researchers to pay attention to the subject.

Here, we briefly highlight certain contributions due to innovative research on the subject. Kishan and Deepa [2] studied the immersion of nanoparticles in micropolar liquid and the stagnation point flow in a porous region. In a nanofluid material flow constrained by a vertical surface, Alim et al. [3] used the Joule heating process. Sheikholeslami et al. [4] conducted research on CuO nanoparticles contained in a heated chamber with a sensodial wall. The heat process in nanomaterial within a micro-channel with a sinusoidal double layer was studied by He et al. [5]. Mahdavi et al. [6] used nanoparticles to identify cooling applications in a hot jet surface. Abdelsalam et al. [7] addressed the thermal repercussions of hybrid nanofluids in blood flow when electro-osmotic forces are present. Nadeem et al. [8] used dual solution simulations to visualise the slip characteristics of nanofluid flow. Khan et al. [9] concentrated on the thermal characteristics of hybrid nanofluids in unsteady flow. Abbas et al. [10] investigated the influence of time-dependent viscosity on nanofluid flow over the Riga surface. Awais et al. [11] used the KKL model to investigate heat transfer in a suspension of nanomaterials containing ($CuO$ and $Al_2O_3$).

Nanomaterials are important because of their high thermal and mechanical properties. The characteristics of the nanoliquids formed by each nanomaterial are considerably altered by these materials. Among nanomaterials, there is a substance known as aluminium alloy, in which aluminium plays a major role. Heat treatable and non-heat treatable alloys are the two main types of aluminium alloys. Aluminium alloys are widely utilised in the construction, testing, and production of spacecraft, aircraft parts, and other structures. Researchers have investigated numerous flow models consisting of aluminium alloys and discovered remarkable thermal transport behaviour due to the improved heat transport features of $AA7072$ and $AA7075$ aluminium alloys. Sandeep and Animasaun [12] reported an examination of heat transfer in nanoliquids consisting of $AA7072$ and $AA7075$ aluminium alloys while considering the impact of varying Lorentz forces. They discovered that nanoliquid made of $AA7075$ alloy is superior in terms of heat transmission to nanoliquid made of $AA7072$. Kandasamy at al. [13] considered the electric field strength for the analysis of heat transport in magnetised $AA7075$ alloys. Tlili et al. [14] investigated three-dimensional heat transfer characteristics in the hybrid colloidal model $AA7072$-$AA7075/Methanol$ under various velocity conditions. They used a numerical approach to the model and described the results in terms of flow regimes.

Since MHD is commonly used in numerous fields, such as the polymer and petroleum industries, a significant amount of thought has been given to the approach of magnetic fields in liquid flow in recent decades. As we know that the pace of cooling is even more essential than in the standard processes, numerous fabrication processes have been used to regulate the rate of cooling for magneto-hydrodynamic liquids. Unifying metals in electric heaters, metal casting, and gem creating are some of the other functions of magneto-hydrodynamics. It also assists in the cooling of the atomic reactor's internal dividers. Magneto-hydrodynamic flows were first sculpted and highly valued in biodesign because they are used in a variety of symptomatic kinds of sickness. In this approach, studying magneto-hydrodynamic flow has a significant impact on several scientific fields. The convective circumstances for MHD Jeffrey flow on an elaborated sheet were examined by Ahmad et al. [15]. Khan et al. [16] studied MHD Falknar-Skan flow through a permeable material with a convective boundary condition. Malik et al. [17] explored MHD hyperbolic flow through an expanded cylinder via numerical methodology, the Kellor-Box method. By assuming magnetic field-dependent viscosity effects, Sheikholeslami et al. [18] described MHD nanofluid flow. The finite element method was used to address this problem.

Flow due to a stretchy surface has risen in prominence among researchers in recent years, owing to its widespread application in industry. Hot rolling, paper production, glass blowing, polymer extrusion, metal extrusion, and crystal growth are only a few of these uses. Crane [19] started flow research with an enlarged sheet. The fluid stream in an enlarged channel was examined by Brady and Acrivos [20]. Researchers discovered that there is a solution for a two-dimensional flow for any given Reynolds number. The movement of fluid past a stretchable cylinder was studied by Wang [21]. By changing the

heat flux, Elbashbeshy [22] investigated heat transfer across a stretchy surface. By using a viscous fluid created by a stretchable surface, Nadeem et al. [23] examined a stagnation-point stream. Different fluid flow over stretched surfaces was inspected by Awan et al. [24]. The production of entropy for MHD Maxwell fluid across a stretchable and penetrable surface was investigated by Jawad et al. [25].

Because of its vast industrial use in nuclear reactor cooling, chemical engineering, geothermal reservoirs, and thermal oil recovery, the chemical reaction effect has gained a tremendous response. Generally, the relationship between mass transfer and chemical reaction is very important, and it can be studied in terms of reactant species deployment and creation at various speeds during nanofluid mass transfer. Bestman et al. [26] conducted ground-breaking work in defining these influences. Mustafa et al. [27] examined a hydromagnetic flow past a radial surface caused by chemical reaction and Arrhenius activation energy. They found that the concentration of a species rises as the activation energy of a chemical process rises. Mohyud-Din et al. [28] looked at how chemical reactions affected convergent/divergent channels. Aleem et al. [29] instigated alternative forms of water based nanofluids, such as titanium-oxide, aluminium-oxide, and copper-oxide, that arose in a porous media after a chemical reaction and Newtonian heating.

The majority of industrial applications necessitate non-Newtonian nanofluids with non-linearly related shear rates and shear stresses. The shear rate is heavily influenced by the timeframe of the shear stress. Thus, coefficients such as viscosity do not fully describe shear stress in such nanofluids. As a result, numerous mathematicians have debated the class of non-Newtonian models, one of which is the Ree-Eyring nanofluid model. Inks, molten polymers, adhesives, paints, organic materials, and other non-Newtonian fluids are some examples. These are used in food industries, drilling rigs, cooling systems, adhesive industries, and so on. Hayat et al. [30] conducted an entropy examination in the flow of Ree-Eyring nanofluid in this respect. Tanveer and Malik [31] investigated the thermal effectiveness of Ree-Eyring nanofluid peristaltic flow. Khan et al. [32] investigated the effect of Lorentz force on the velocity of a Ree-Eyring nanofluid flow past a paraboloid surface. Al-Mdallal et al. [33] investigated the thermal properties of *Cu-Water* nanofluid under the sway of radiation. Purna et al. [34] used the Darcy-Frochheimer law to examine the flow of Ree-Eyring nanofluid on a porous plate inclined at an angle, as well as the impact of the chemical reaction. Some recent studies about nanofluids and heat transfer properties are mentioned in Refs. [35–38].

The main goal of this article is to study the existence of a magnetic dipole and the Koo-Kleinstreuer model using different alloys over a stretching sheet. The Cattaneo-Christov model is used to calculate heat transfer in a two-dimensional flow of Ree-Eyring nanofluid across a stretching sheet. The mathematical formulation is created in the following section, utilising fluid flow assumptions. By applying appropriate similarity transformations, the physical flow phenomenon is represented and then translated into a non-dimensional form. The HAM is used to arrive at a solution. Through graphical demonstrations, the influences of a few key parameters on the temperature, velocity fields, and concentration profile are highlighted.

## 2. Mathematical Model and Formulation

We consider a Ree-Eyring nanofluid flow in a two-dimensional laminar boundary layer with the influence of magnetic dipoles. Furthermore, the Cattaneo-Christov heat flow model is used to analyse heat transmission. The magnetic dipole is located under the sheet, whereas the electrically non-conductive and incompressible Ree-Eyring nanofluid is located above the sheet in the half-space $y > 0$. Figure 1 depicts the flow geometry. By assuming two conflicting and comparable forces along the *x-axis*, the sheet is stretched at a proportional rate to the distance between it and the fixed origin $x = 0$. The dipole centre is located on the *y-axis* below the *x-axis*. It has a powerful magnetic field directed in the positive *x-direction*, which increases the magnetic field's intensity enough to feed the

Ree-Eyring nanofluid. The stretched sheet is kept at a temperature $T_w$ underneath Curie's temperature $T_c$, although the far-flung liquid elements are thought to be at $T = T_c$.

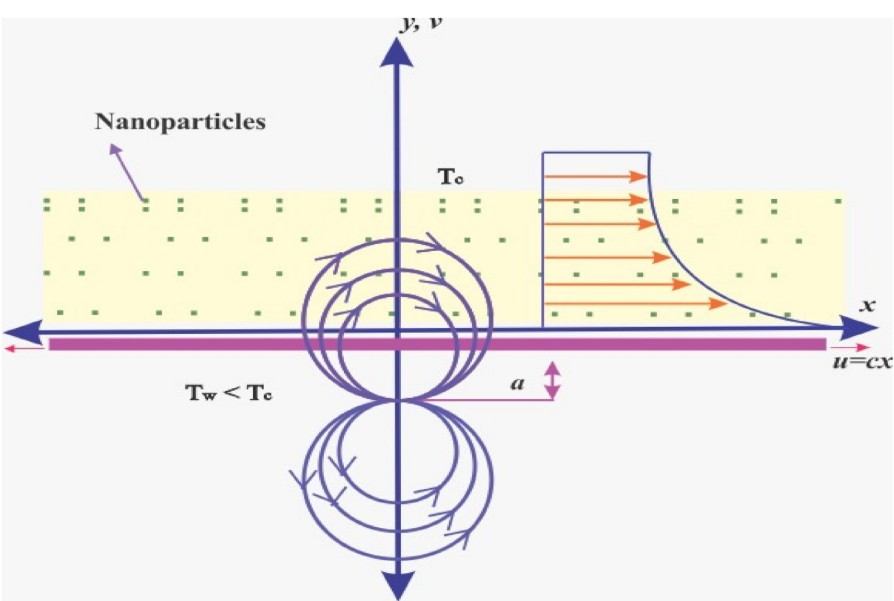

**Figure 1.** Physical description flow geometry.

Using the Koo-Kleinstreuer model, the efficacious $k_{nf}$ of nanofluids can be displayed as [11]

$$k_{nf} = k_{static} + k_{Brownian}$$

where

$$k_{static} = \frac{k_f\left(k_s + 2k_f\right) + 2\phi\left(k_s - k_f\right)}{k_f\left(k_s + 2k_f\right) - \phi\left(k_s - k_f\right)} \cdot k_{Brownian} = 5 \times 10^4 \gamma \varnothing \left(\rho C_p\right)_f \sqrt{\frac{k_\beta T}{2\rho_p r_p}} \Gamma(T, \phi),$$

where $k_\beta = 1.38 \times 10^{-23} \mathrm{m^2 kgs^{-2} k^{-1}}$ is the Boltzmann physical constant and $r_p$ is the nanoparticle radius.

Particularly,

$\gamma = 0.0137(100\phi)^{-0.8229}$, where $\phi < 1\%$;

$\gamma = 0.0011(100\phi)^{-0.7272}$, where $\phi > 1\%$;

$0.01 < \phi < 0.04$ $300K < T < 325K$.

Taking into account $\mu_{nf}$ reliance on particle volume fraction,

$$\mu_{nf} = \mu_{static} + \mu_{Brownian}$$

where

$$\text{and } \mu_{static} = \mu_f(1 - \phi)^{-2.5} \text{ and } \mu_{Brownian} = \frac{k_{Brownian}}{k_f}\frac{\mu_f}{Pr_f}$$

The governed equation is formulated as follows [32,36,37]:

$$\frac{\partial u}{\partial x} + \frac{\partial v}{\partial y} = 0 \tag{1}$$

$$u\frac{\partial u}{\partial x} + v\frac{\partial u}{\partial y} = \frac{1}{\rho_{nf}}\left(\frac{1}{\beta_1\epsilon} + \mu_{nf}\right)\left(2\frac{\partial^2 u}{\partial x^2} + \frac{\partial^2 v}{\partial x \partial y} + \frac{\partial^2 u}{\partial y^2}\right) + \frac{\mu_0 M}{\rho_{nf}}\frac{\partial H}{\partial x} \tag{2}$$

$$u\frac{\partial T}{\partial x} + v\frac{\partial T}{\partial y} + \lambda_2 \Omega_E = \frac{k_{nf}}{(\rho C_p)_{nf}}\left(\frac{\partial^2 T}{\partial y^2} + \frac{\partial^2 T}{\partial x^2}\right) + \frac{1}{(\rho C_p)_{nf}}\left(\frac{1}{\beta_1 \epsilon} + \mu_{nf}\right)\left(\frac{\partial u}{\partial y}\right)^2$$
$$- \frac{\mu_0 T}{(\rho C_p)_{nf}}\frac{\partial M}{\partial T}\left(u\frac{\partial H}{\partial x} + v\frac{\partial H}{\partial y}\right) \tag{3}$$

$$u\frac{\partial C}{\partial x} + v\frac{\partial C}{\partial y} = D_{nf}\left(\frac{\partial^2 T}{\partial y^2} + \frac{\partial^2 T}{\partial x^2}\right) - k_r(C - C_c) \tag{4}$$

In the above equation, the term $\Omega_E$ is defined as

$$\Omega_E = u\frac{\partial u}{\partial x}\frac{\partial T}{\partial x} + v\frac{\partial v}{\partial y}\frac{\partial T}{\partial y} + u^2\frac{\partial^2 T}{\partial x^2} + v^2\frac{\partial^2 T}{\partial y^2} + 2uv\frac{\partial^2 T}{\partial x \partial y} + u\frac{\partial v}{\partial x}\frac{\partial T}{\partial y} + v\frac{\partial u}{\partial y}\frac{\partial T}{\partial y} \tag{5}$$

The associated boundary limitations are as follows:

$$\left.\begin{array}{c} u = cx,\ v = 0,\ T = T_w,\ C = C_w\ at\ y = 0 \\ u \to 0,\ T \to T_c,\ C \to C_c\ at\ y \to \infty \end{array}\right\} \tag{6}$$

The magnetic field affects the presumed liquid flow due to the magnetic dipole, and its magnetic scalar potential is given by

$$\phi_1 = \frac{x}{(y+a)^2 + x^2}\frac{\gamma}{2\pi} \tag{7}$$

$$H_y = -\frac{\partial \phi_1}{\partial y} = \frac{2(y+a)x}{((y+a)^2 + x^2)^2}\frac{\gamma}{2\pi},\ H_y = -\frac{\partial \phi_1}{\partial x} = -\frac{(y+a)^2 - x^2}{((y+a)^2 + x^2)^2}\frac{\gamma}{2\pi} \tag{8}$$

where

$$H = \left[\left(\frac{\partial \phi_1}{\partial y}\right)^2 + \left(\frac{\partial \phi_1}{\partial x}\right)^2\right]^{1/2} \tag{9}$$

We attain that

$$\frac{\partial H}{\partial y} = \left[\frac{4x^2}{(y+a)^5} - \frac{2}{(y+a)^3}\right]\frac{\gamma}{2\pi},\ \frac{\partial H}{\partial x} = \left[-\frac{2x}{(y+a)^4}\right]\frac{\gamma}{2\pi} \tag{10}$$

Supposing that the applied field $H$ is strong enough to saturate the supposed fluid and that the linear equation approximates the variance of magnetisation $M$ with temperature $T$,

$$M = K(T_c - T) \tag{11}$$

The following are some of the similarities:

$$(\eta, \xi) = \sqrt{\frac{c}{v_f}}(y, x),\ \psi(\eta, \xi) = \left(\frac{\mu_f}{\rho_f}\right)\xi f(\eta)$$
$$T = T_c - (T_c - T_w)\theta(\eta, \xi) = T_c - (T_c - T_w)[\theta_1(\eta) + \xi^2 \theta_2(\eta)] \tag{12}$$

The stream function $\psi$ is given below:

$$u = \frac{\partial \psi}{\partial y} = cxf'(\eta),\ v = -\frac{\partial \psi}{\partial x} = -\sqrt{cv_f}f(\eta) \tag{13}$$

The continuity equation is easily satisfied, while the momentum, thermal equations, and mass transfer are transferred to the relating set of ODEs:

$$\left(\varepsilon_2 We + \varepsilon_1 + \frac{k_{Brownian}}{k_f Pr_f \varepsilon_2}\right)f''' - \varepsilon_2 \frac{2\beta\theta_1}{(\eta + \alpha)^4} + ff'' - f'^2 = 0 \tag{14}$$

$$\varepsilon_3 \frac{k_{nf}}{k_f}\frac{1}{Pr}(\theta_1'' + 2\theta_2) + f\theta_1' + \varepsilon_3 \frac{1}{Pr}\frac{2\lambda\beta}{(\eta + \alpha)^4}f(\theta_1 - \varepsilon) - \delta_e\left(f^2\theta_1'' + ff'\theta_1'\right) = 0 \tag{15}$$

$$\varepsilon_3 \frac{k_{nf}}{k_f} \frac{1}{Pr} \theta_2'' + f\theta_2 \frac{\lambda}{Pr} \left( \varepsilon_2 We + \varepsilon_1 + \frac{k_{Brownian}}{k_f Pr_f \varepsilon_2} \right) f'^2 - \varepsilon_3 \frac{\lambda\beta(\theta_1 - \delta)}{Pr} \left[ \frac{4f}{(\eta + \alpha)^5} + \frac{2f'}{(\eta + \alpha)^4} \right] = 0 \quad (16)$$
$$+ \varepsilon_3 \frac{1}{Pr} \frac{2\lambda\beta}{(\eta + \alpha)^3} f\theta_2 - \delta_e \left( f^2 \theta_2'' - 3ff'\theta_2' - 2ff''\theta_2 + 4f'^2 \theta_2 \right)$$

$$(1 - \phi)^{2.5} \frac{1}{Sc} \left( \chi_1'' + 2\chi_2 \right) + f\chi_1' - \sigma\chi_1 = 0 \quad (17)$$

$$(1 - \phi)^{2.5} \frac{1}{Sc} \chi_2'' + f\chi_2' - 2\chi_2 f' - \sigma\chi_2 = 0 \quad (18)$$

where

$$\varepsilon_1 = \frac{1}{(1 - \phi)^{2.5} \left( 1 - \phi + \phi \frac{\rho_s}{\rho_f} \right)}, \ \varepsilon_2 = \frac{1}{\left( 1 - \phi + \phi \frac{\rho_s}{\rho_f} \right)}, \ \varepsilon_3 = \frac{1}{\left( 1 - \phi + \phi \frac{(\rho C_p)_s}{(\rho C_p)_f} \right)} \quad (19)$$

Reduced conditions:

$$\left. \begin{array}{l} f(0) = 0, \ f'(0) = 1, \ \theta_1(0) = 1, \ \theta_2(0) = 0, \ \chi_1(0) = 1, \ \chi_2(0) = 0 \\ f'(\infty) \to 0, \ \theta_1(\infty) \to 0, \ \theta_2(\infty) \to 0, \ \chi_1(\infty) \to 0, \ \chi_2(\infty) \to 0 \end{array} \right\} \quad (20)$$

where

$$\alpha = \sqrt{\frac{c}{v_f}} a, \ \beta = \mu_0 K \frac{\gamma\rho_f}{2\pi\mu_f^2} (T_c - T_w), \ We = \frac{1}{\beta_1 \epsilon \mu_f}, \ \delta_e = c\lambda_2, \ \delta = \frac{T_c}{(T_c - T_w)} \quad (21)$$
$$\lambda = \frac{c\mu_f^2}{k_f \rho_f (T_c - T_w)}, \ Pr = \frac{\mu_f C_p}{k_f}, \ \sigma = \frac{k_r}{c}, \ Sc = \frac{v_f}{D_f}, \ Re = \frac{cx^2}{v_f}$$

The definitions of the quantities of physical interests are as follows:

$$C_{f_x} = \frac{-2 \left( \frac{1}{\beta\epsilon} + \mu_{nf} \right) \left( \frac{\partial^2 u}{\partial x^2} \right)_{y=0}}{\rho(cx)^2}, \quad Nu_x = \frac{-xk_{nf} \left( \frac{\partial T}{\partial y} \right)_{y=0}}{(T_w - T_c)}, \quad (22)$$
$$Sh_x = \frac{-x \left( \frac{\partial C}{\partial y} \right)_{y=0}}{(C_w - C_c)}$$

The quantities of physical interest corresponding to Equations (12) and (13) transform the following equations:

$$\sqrt{Re_x} C_{f_x} = -\frac{2(1 + We)}{(1 - \phi)^{2.5}} f''(0) \quad (23)$$

$$(Re_x)^{-1/2} Nu_x = -\frac{k_{nf}}{k_f} \left( \theta_1'(0) + \xi^2 \theta_2'(0) \right) \quad (24)$$

$$(Re_x)^{-1/2} Sh_x = -(1 - \phi)^{2.5} \left( \chi_1'(0) + \xi^2 \chi_2'(0) \right) \quad (25)$$

## 3. Solution Method and Details

In order to solve Equations (14–18) under the boundary conditions (19, 20), we use the Homotopy Analysis Method (HAM) with the following procedure. The solutions with the auxiliary parameters $\hbar$ adjust and control the convergence of the solutions.

The initial guesses are selected as follows:

$$f_0(\eta) = (1 - e^{-\eta}), \ \theta_{1,0}(\eta) = e^{-\eta}, \ \theta_{2,0}(\eta) = \eta e^{-\eta}, \ \chi_{1,0}(\eta) = e^{-\eta}, \ \chi_{2,0}(\eta) \quad (26)$$

The linear operators are taken as $L_f, \ L_{\theta_1}, \ L_{\theta_2}, \ L_{\chi_1}, \ L_{\chi_2}$

$$L_f(f) = f''' - f', \ L_{\theta_1}(\theta_1) = \theta_1'' - \theta_1, \ L_{\theta_2}(\theta_2) = \theta_2'' - \theta_2 \quad (27)$$
$$L_{\chi_1}(\chi_1) = \chi_1'' - \chi_1, \ L_{\chi_2}(\chi_2) = \chi_2'' - \chi_2$$

which have the following properties:

$$L_f(c_1 + c_2 e^{-\eta} + c_3 e^{\eta}) = 0, \ L_{\theta_1}(c_4 e^{\eta} + c_5 e^{-\eta}) = 0$$
$$L_{\theta_2}(c_6 e^{\eta} + c_7 e^{-\eta}) = 0, \ L_{\chi_1}(c_8 e^{-\eta} + c_9 e^{\eta}) = 0, \ L_{\chi_1}(c_{10} e^{-\eta} + c_{11} e^{\eta}) = 0 \quad (28)$$

where $c_i (i = 1 - 11)$ are the constants in general solution:

The resultant non-linear operatives $N_f$, $N_{\theta_1}$, $N_{\theta_2}$, $N_{\chi_1}$, $N_{\chi_2}$ are given as

$$N_f[f(\eta; p), \ \theta_1(\eta; p)] = \left( \varepsilon_2 We + \frac{k_{Brownian}}{k_f Pr_f \varepsilon_2} \right) \frac{\partial^3 f(\eta; p)}{\partial \eta^3} - \varepsilon_2 \frac{2\beta}{(\eta + \alpha)^4} \theta_1(\eta; p) - \left( \frac{\partial f(\eta; p)}{\partial \eta} \right)^2 + f(\eta; p) \frac{\partial^2 f(\eta; p)}{\partial \eta^2} \quad (29)$$

$$N_{\theta_1}[f(\eta; p), \ \theta_1(\eta; p), \ \theta_2(\eta; p)] = \varepsilon_2 \frac{k_{nf}}{k_f} \frac{1}{Pr} \left( \frac{\partial^2 \theta_1(\eta; p)}{\partial \eta^2} + 2\theta_2(\eta; p) \right) +$$
$$f(\eta; p) \frac{\partial \theta_1(\eta; p)}{\partial \eta} + \varepsilon_2 \frac{1}{Pr} \frac{2\lambda}{(\eta + \alpha)^4} (f(\eta; p)\theta_1(\eta; p) - \varepsilon f(\eta; p)) - \quad (30)$$
$$\delta_e \left( (f(\eta; p))^2 \frac{\partial^2 \theta_1(\eta; p)}{\partial \eta^2} + f(\eta; p) \frac{\partial f(\eta; p)}{\partial \eta} \frac{\partial \theta_1(\eta; p)}{\partial \eta} \right)$$

$$N_{\theta_2}[f(\eta; p), \theta_1(\eta; p), \theta_2(\eta; p)] = \varepsilon_2 \frac{k_{nf}}{k_f} \frac{1}{Pr} \frac{\partial^2 \theta_2(\eta; p)}{\partial \eta^2} +$$
$$\frac{\lambda}{Pr} \left( \varepsilon_2 We + \varepsilon_1 \frac{k_{Brownian}}{k_f Pr_f \varepsilon_2} \right) f(\eta; p) \left( \frac{\partial f(\eta; p)}{\partial \eta} \right)^2 \theta_1(\eta; p) + \varepsilon_2 \frac{2\lambda}{Pr} \frac{1}{(\eta + \alpha)^3} f(\eta; p)\theta_2(\eta; p)$$
$$- \varepsilon_2 \frac{\lambda\beta}{Pr} \left[ \frac{4}{(\eta + \alpha)^5} f(\eta; p) + \frac{2}{(\eta + \alpha)^4} \frac{\partial f(\eta; p)}{\partial \eta} \right] \quad (31)$$
$$\delta_e \left( \begin{array}{c} (f(\eta; p))^2 \frac{\partial^2 \theta_2(\eta; p)}{\partial \eta^2} - 3f(\eta; p) \frac{\partial f(\eta; p)}{\partial \eta} \frac{\partial \theta_2(\eta; p)}{\partial \eta} \\ -f(\eta; p) \frac{\partial^2 f(\eta; p)}{\partial \eta^2} \theta_2(\eta; p) + 4 \left( \frac{\partial f(\eta; p)}{\partial \eta} \right)^2 \theta_2(\eta; p) \end{array} \right)$$

$$N_{\chi_1}[f(\eta; p), \ \chi_1(\eta; p), \ \chi_2(\eta; p)] = (1 - \phi)^{2.5} \frac{1}{Sc} \left( \frac{\partial^2 \chi_1(\eta; p)}{\partial \eta^2} + 2\chi_2(\eta; p) \right) +$$
$$f(\eta; p) \frac{\partial \chi_1(\eta; p)}{\partial \eta} - \sigma \chi_1(\eta; p) \quad (32)$$

$$N_{\chi_2}[f(\eta; p), \ \chi_2(\eta; p)] = (1 - \phi)^{2.5} \frac{1}{Sc} \frac{\partial^2 \chi_1(\eta; p)}{\partial \eta^2} + f(\eta; p) \frac{\partial \chi_2(\eta; p)}{\partial \eta}$$
$$-2 \frac{\partial f(\eta; p)}{\partial \eta} \chi_2(\eta; p) - \sigma \chi_2(\eta; p) \quad (33)$$

The basic idea of the HAM is described in [1–7]; the zeroth-order problems from Equations (14)–(18) are

$$(1 - p)L_f[f(\eta; p) - f_0(\eta)] = p\hbar_f N_f[f(\eta; p), \ \theta_1(\eta; p)] \quad (34)$$

$$(1 - p)L_{\theta_1}[\theta_1(\eta; p) - \theta_{1,0}(\eta)] = p\hbar_{\theta_1} N_{\theta_1}[f(\eta; p), \ \theta_1(\eta; p), \ \theta_2(\eta; p)] \quad (35)$$

$$(1 - p)L_{\theta_2}[\theta_2(\eta; p) - \theta_{2,0}(\eta)] = p\hbar_{\theta_2} N_{\theta_2}[f(\eta; p), \ \theta_1(\eta; p), \ \theta_2(\eta; p)] \quad (36)$$

$$(1 - p)L_{\chi_1}[\chi_1(\eta; p) - \chi_{1,0}(\eta)] = p\hbar_{\chi_1} N_{\chi_1}[f(\eta; p), \ \chi_1(\eta; p), \ \chi_2(\eta; p)] \quad (37)$$

$$(1 - p)L_{\chi_1}[\chi_2(\eta; p) - \chi_{2,0}(\eta)] = p\hbar_{\chi_2} N_{\chi_2}[f(\eta; p), \ \chi_2(\eta; p)] \quad (38)$$

The equivalent boundary conditions are

$$f(\eta; p)|_{\eta=0} = 0, \ \left. \frac{\partial f(\eta; p)}{\partial \eta} \right|_{\eta=0} = 1, \ \left. \frac{\partial f(\eta; p)}{\partial \eta} \right|_{\eta \to \infty} = 0$$
$$\theta_1(\eta; p)|_{\eta=0} = 1, \ \theta_1(\eta; p)|_{\eta \to \infty} = 0$$
$$\theta_2(\eta; p)|_{\eta=0} = 0, \ \theta_2(\eta; p)|_{\eta \to \infty} = 0 \quad (39)$$
$$\chi_1(\eta; p)|_{\eta=0} = 1, \ \chi_1(\eta; p)|_{\eta \to \infty} = 0$$
$$\chi_2(\eta; p)|_{\eta=0} = 0, \ \chi_2(\eta; p)|_{\eta \to \infty} = 0$$

where $p \in [0,1]$ is the imbedding parameter; $\hbar_f$, $\hbar_{\theta_1}$, $\hbar_{\theta_2}$, $\hbar_{\chi_1}$, $\hbar_{\chi_2}$ are used to control the convergence of the solution. When $p = 0$ and $p = 1$, we have

$$f(\eta;1) = f(\eta), \ \theta_1(\eta;1) = \theta_1(\eta), \ \theta_2(\eta;1) = \theta_2(\eta), \ \chi_1(\eta;1) = \chi_1(\eta), \ \chi_2(\eta;1) = \chi_2(\eta) \qquad (40)$$

Expanding $f(\eta;p)$, $\theta_1(\eta;p)$, $\theta_2(\eta;p)$, $\chi_1(\eta;p)$, $\chi_2(\eta;p)$ in Taylor's series about $p = 0$

$$
\begin{aligned}
f(\eta;p) &= f_0(\eta) + \sum_{m=1}^{\infty} f_m(\eta) p^m \\
\theta_1(\eta;p) &= \theta_{1,\,0}(\eta) + \sum_{m=1}^{\infty} \theta_{1,m}(\eta) p^m \\
\theta_2(\eta;p) &= \theta_{2,\,0}(\eta) + \sum_{m=1}^{\infty} \theta_{2,m}(\eta) p^m \\
\chi_1(\eta;p) &= \chi_{1,\,0}(\eta) + \sum_{m=1}^{\infty} \chi_{1,m}(\eta) p^m \\
\chi_2(\eta;p) &= \chi_{2,\,0}(\eta) + \sum_{m=1}^{\infty} \chi_{2,m}(\eta) p^m
\end{aligned}
\qquad (41)
$$

where

$$
f_m(\eta) = \frac{1}{m!} \left. \frac{\partial f(\eta;p)}{\partial \eta} \right|_{p=0}, \ \theta_{1,\,m}(\eta) = \frac{1}{m!} \left. \frac{\partial \theta_1(\eta;p)}{\partial \eta} \right|_{p=0}
$$

$$
\theta_{2,\,m}(\eta) = \frac{1}{m!} \left. \frac{\partial \theta_2(\eta;p)}{\partial \eta} \right|_{p=0}, \ \chi_{1,\,m}(\eta) = \frac{1}{m!} \left. \frac{\partial \chi_1(\eta;p)}{\partial \eta} \right|_{p=0}, \ \chi_{2,\,m}(\eta) = \frac{1}{m!} \left. \frac{\partial \chi_2(\eta;p)}{\partial \eta} \right|_{p=0}
$$

$$(42)$$

The secondary constraints $\hbar_f$, $\hbar_{\theta_1}$, $\hbar_{\theta_2}$, $\hbar_{\chi_1}$, $\hbar_{\chi_2}$ are chosen in such a way that the series (40) converges at $p = 1$; switching $p = 1$ in (40), we obtain

$$
\begin{aligned}
f(\eta) &= f_0(\eta) + \sum_{m=1}^{\infty} f_m(\eta) \\
\theta_1(\eta) &= \theta_{1,\,0}(\eta) + \sum_{m=1}^{\infty} \theta_{1,m}(\eta) \\
\theta_2(\eta) &= \theta_{2,\,0}(\eta) + \sum_{m=1}^{\infty} \theta_{2,m}(\eta) \\
\chi_1(\eta) &= \chi_{1,\,0}(\eta) + \sum_{m=1}^{\infty} \chi_{1,m}(\eta) \\
\chi_2(\eta) &= \chi_{2,\,0}(\eta) + \sum_{m=1}^{\infty} \chi_{2,m}(\eta)
\end{aligned}
\qquad (43)
$$

The $m^{th}$-*order* problem satisfies the following:

$$
\begin{aligned}
L_f[f_m(\eta) - \chi_m f_{m-1}(\eta)] &= \hbar_f R_m^f(\eta) \\
L_{\theta_1}[\theta_{1,m}(\eta) - \chi_m \theta_{1,m-1}(\eta)] &= \hbar_{\theta_1} R_m^{\theta_1}(\eta) \\
L_{\theta_2}[\theta_{2,m}(\eta) - \chi_m \theta_{2,m-1}(\eta)] &= \hbar_{\theta_2} R_m^{\theta_2}(\eta) \\
L_{\chi_1}[\chi_{1,m}(\eta) - \chi_m \chi_{1,m-1}(\eta)] &= \hbar_{\chi_1} R_m^{\chi_1}(\eta) \\
L_{\chi_2}[\chi_{2,m}(\eta) - \chi_m \chi_{2,m-1}(\eta)] &= \hbar_{\chi_2} R_m^{\chi_2}(\eta)
\end{aligned}
\qquad (44)
$$

The corresponding boundary conditions are as follows:

$$
\begin{aligned}
f_m(0) = f_m'(0) = \theta_{1,\,m}'(0) = \theta_{2,\,m}'(0) = \chi_{1,\,m}(0) = \chi_{2,\,m}(0) &= 0 \\
f_m'(\infty) = \theta_{1,\,m}(\infty) = \theta_{2,\,m}(\infty) = \chi_{1,\,m}(\infty) = \chi_{2,\,m}(\infty) &= 0
\end{aligned}
\qquad (45)
$$

Here

$$
R_m^f(\eta) = \left( \varepsilon_2 We + \frac{k_{Brownian}}{k_f Pr_f \varepsilon_2} \right) f_{m-1}''' - \varepsilon_2 \frac{2\beta}{(\eta+\alpha)^4} \theta_{1,\,m-1} - \frac{\partial^3 f(\eta;p)}{\partial \eta^3} -
$$

$$
\varepsilon_2 \frac{2\beta}{(\eta+\alpha)^4} \theta_1(\eta;p) - \sum_{k=0}^{m-1} f_{m-1}' f_k' + \sum_{k=0}^{m-1} f_{m-1-k} f_k''
$$

$$(46)$$

$$R_m^{\theta_1}(\eta) = \varepsilon_2 \frac{k_{nf}}{k_f} \frac{1}{Pr} \left( \theta_{1,\,m-1}'' + 2\theta_{2,\,m-1} \right) + \varepsilon_2 \frac{1}{Pr} \frac{2\lambda}{(\eta+\alpha)^4} \left( \sum_{k=0}^{m-1} f_{m-1-k}\theta_{1,\,k} - \varepsilon f_{m-1} \right)$$
$$+ \sum_{k=0}^{m-1} f_{m-1-k}\theta_{1,\,k}' - \delta_e \left[ \sum_{k=0}^{m-1} f_{m-1-k} \sum_{i=0}^{k} f_{k-i}\theta_{1,i}' + \sum_{k=0}^{m-1} f_{m-1-k} \sum_{i=0}^{k} f_{k-i}'\theta_{1,\,i} \right] \tag{47}$$

$$R_m^{\theta_2}(\eta) = \varepsilon_2 \frac{k_{nf}}{k_f} \frac{1}{Pr} \theta_{2,m-1}'' + \frac{\lambda}{Pr} \left( \varepsilon_2 We + \varepsilon_1 \frac{k_{Brownian}}{k_f Pr_f \varepsilon_2} \right) \sum_{k=0}^{m-1} f_{m-1-k} \sum_{i=0}^{k} f_k' \sum_{p=0}^{i} f_{i-p}'\theta_{1,p}$$
$$- \varepsilon_2 \frac{\lambda\beta}{Pr} \left[ \frac{4}{(\eta+\alpha)^5} f_{m-1} + \frac{2}{(\eta+\alpha)^4} f_{m-1}' \right] + \varepsilon_2 \frac{2\lambda\beta}{Pr} \frac{1}{(\eta+\alpha)^3} \sum_{k=0}^{m-1} f_{m-1-k}\theta_{2,k}$$
$$- \delta_e \left[ \begin{array}{c} \sum_{k=0}^{m-1} f_{m-1-k} \sum_{i=0}^{k} f_{k-i}\theta_{2,i}'' - 3\sum_{k=0}^{m-1} f_{m-1-k} \sum_{i=0}^{k} f_{k-i}'\theta_{2,i} - \\ \sum_{k=0}^{m-1} f_{m-1-k} \sum_{i=0}^{k} f_{k-i}''\theta_{2,i} + 4\sum_{k=0}^{m-1} f_{m-1-k}' \sum_{i=0}^{k} f_{k-i}'\theta_{2,i} \end{array} \right] \tag{48}$$

$$R_m^{\chi_1}(\eta) = (1-\phi)^{2.5} \frac{1}{Sc} \left( \chi_{1,\,m-1}'' + 2\chi_{2,\,m-1} \right) + \sum_{k=0}^{m-1} f_{m-1-k}\chi_{1,k}' - \sigma\chi_{1,\,m-1} \tag{49}$$

$$R_m^{\chi_2}(\eta) = (1-\phi)^{2.5} \frac{1}{Sc} \chi_{2,\,m-1}'' + \sum_{k=0}^{m-1} f_{m-1-k}\chi_{2,k}' - \sum_{k=0}^{m-1} f_{m-1-k}'\chi_{2,k} - \sigma\chi_{2,\,m-1} \tag{50}$$

where

$$\chi_m = \begin{cases} 0, & if\ p \leq 1 \\ 1, & if\ p > 1 \end{cases} \tag{51}$$

*Validation and Comparison*

Table 1 shows the physical properties of nanofluid. A comparison of the validation of the results for the velocity, temperature, and concentration fields using the Homotopy Analysis Method and a numerical (ND-solve) method are shown in Tables 2–4 and in Figures 2–4. From these tables and figures, it can be observed that the results of both methods are in good agreement.

**Table 1.** The base fluids' and nanoparticles' material properties.

| Fluids | $\rho$ (kg/m$^3$) | $C_p$ (J/kg·K) | $k$ (W/m·K) |
|---|---|---|---|
| H$_2$O | 997.1 | 4179 | 0.613 |
| *AA7072* | 2810 | 960 | 173 |
| *AA7075* | 2720 | 893 | 222 |

**Table 2.** Comparison table for HAM solution and numerical method for velocity field and their results.

| $\eta$ | HAM Solution | Numerical Solution | Absolute Error |
|---|---|---|---|
| 0.0 | 1.000000 | 1.000000 | 0.000000 |
| 0.5 | 0.672325 | 0.672090 | 0.000235 |
| 1.0 | 0.450868 | 0.450368 | 0.000501 |
| 1.5 | 0.300635 | 0.300102 | 0.000532 |
| 2.0 | 0.198485 | 0.198011 | 0.000474 |
| 2.5 | 0.128841 | 0.128458 | 0.000383 |
| 3.0 | 0.081224 | 0.080938 | 0.000286 |
| 3.5 | 0.048575 | 0.048379 | 0.000196 |
| 4.0 | 0.026132 | 0.026014 | 0.000118 |
| 4.5 | 0.010669 | 0.010617 | 0.000053 |
| 5.0 | $8.649750 \times 10^{-8}$ | $1.697710 \times 10^{-8}$ | $6.952040 \times 10^{-8}$ |

**Table 3.** Comparison table for HAM solution and numerical method for temperature field and their results.

| $\eta$ | HAM Solution | Numerical Solution | Absolute Error |
|---|---|---|---|
| 0.0 | 1.000000 | 1.000000 | $2.775560'' \times 10^{-15}$ |
| 0.5 | 1.422530 | 1.422300 | 0.000240 |
| 1.0 | 1.520240 | 1.519810 | 0.000432 |
| 1.5 | 1.422380 | 1.421830 | 0.000551 |
| 2.0 | 1.222520 | 1.221930 | 0.000592 |
| 2.5 | 0.982008 | 0.981442 | 0.000566 |
| 3.0 | 0.737851 | 0.737362 | 0.000489 |
| 3.5 | 0.510493 | 0.510113 | 0.000380 |
| 4.0 | 0.309869 | 0.309615 | 0.000254 |
| 4.5 | 0.139648 | 0.139524 | 0.000124 |
| 5.0 | $-2.16757'' \times 10^{-7}$ | $7.061450'' \times 10^{-8}$ | $2.873720'' \times 10^{-7}$ |

**Table 4.** Comparison table for HAM solution and numerical method for concentration field and their results.

| $\eta$ | HAM Solution | Numerical Solution | Absolute Error |
|---|---|---|---|
| 0.0 | 1.000000 | 1.000000 | 0.000000 |
| 0.5 | 0.662316 | 0.662225 | 0.000090 |
| 1.0 | 0.449273 | 0.449152 | 0.000121 |
| 1.5 | 0.309253 | 0.309134 | 0.000119 |
| 2.0 | 0.213854 | 0.213750 | 0.000104 |
| 2.5 | 0.146837 | 0.146752 | 0.000085 |
| 3.0 | 0.098537 | 0.098472 | 0.000066 |
| 3.5 | 0.062965 | 0.062918 | 0.000047 |
| 4.0 | 0.036264 | 0.036234 | 0.000030 |
| 4.5 | 0.015864 | 0.015850 | 0.000014 |
| 5.0 | $1.648660'' \times 10^{-7}$ | $5.280260'' \times 10^{-8}$ | $1.120640'' \times 10^{-7}$ |

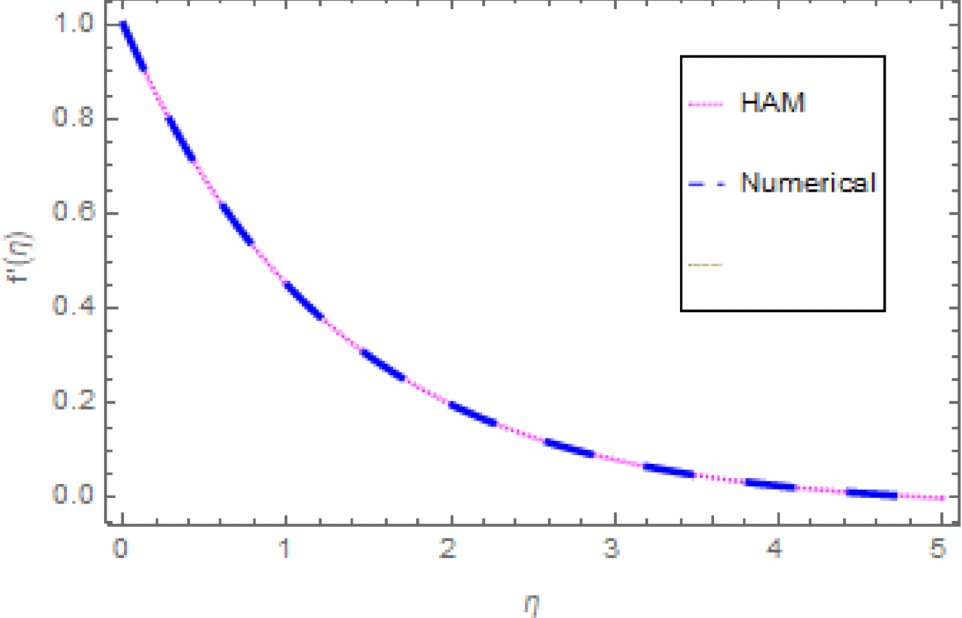

**Figure 2.** Comparison graph for velocity profile.

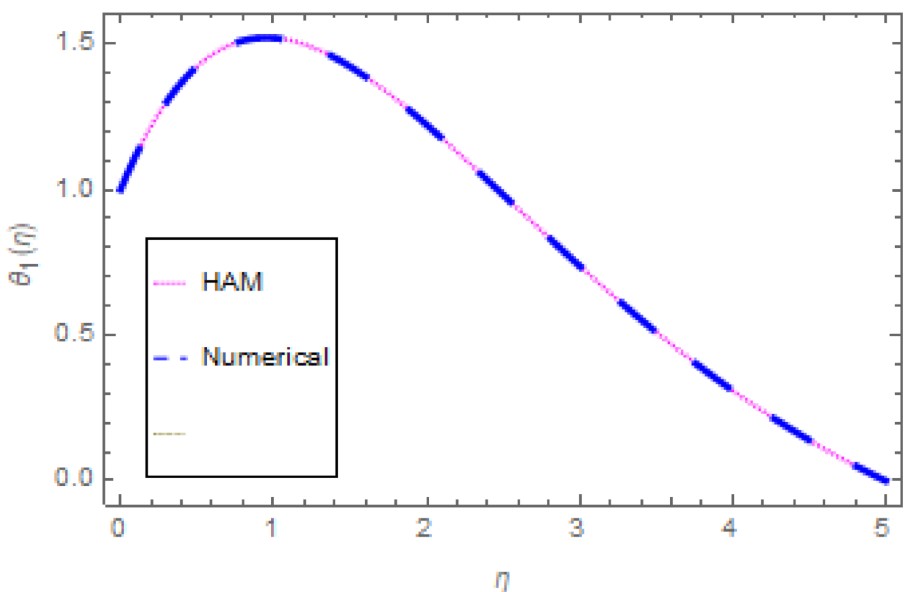

**Figure 3.** Comparison graph for temperature profile.

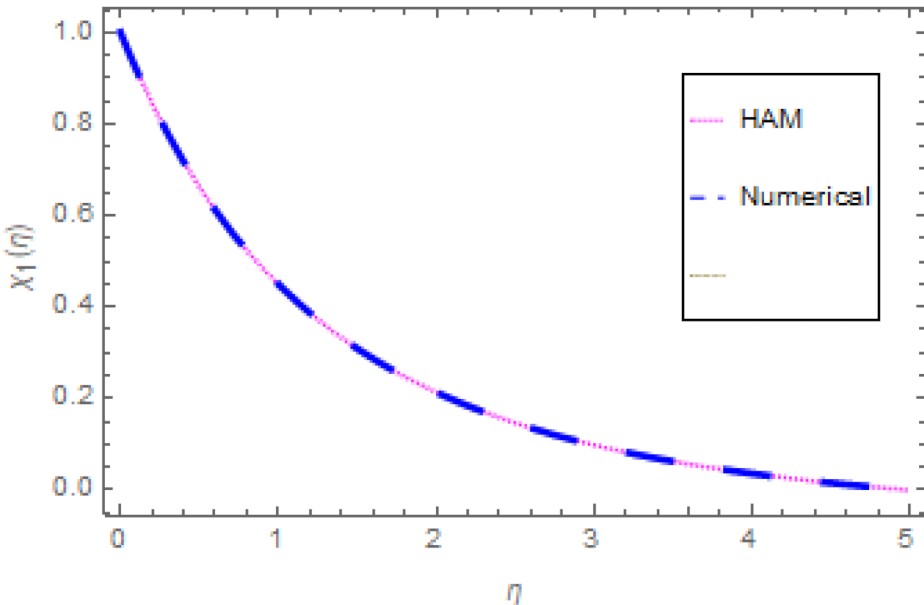

**Figure 4.** Comparison graph for concentration profile.

## 4. Results and Discussion

This section uses plotted figures to discuss the physical aspects of flow, thermal field changes, and concentration profile, and to explain physical interpretations triggered by the dominant dimensionless factors. The HAM is used to solve shortened ODEs numerically. Two diverse cases, namely, the $AA7072$ alloy and the $AA7075$ alloy, are well-thought-out in this analysis. Graphs are used to explain the effects of various specifications on $f'(\eta)$, $\theta_1(\eta)$, and $\chi_1(\eta)$, such as the Ree-Eyring fluid parameter, the ferromagnetic interaction parameter, the Schmidt number, the Prandtl number, and the reaction rate parameter. Additionally, skin friction and the Nusselt number are illustrated graphically.

Figure 5 shows the change in $f'(\eta)$ of both alloys $AA7072$ and $AA7075$ as $\beta$ changes. In this case, increasing $\beta$ lowers the $f'(\eta)$ of both alloys. This means that a large mass flux can reduce the velocity of the liquid on the surface. The occurrence of $\beta$ and Curie temperature in this circumstance is crucial to consider the ferromagnetic stimulus on the flow, which upsurges liquid viscosity and diminishes the velocity gradient. Physically, when the

magnetic influence is absent, the fluid velocity upsurges. The magnetic dipole effect, on the other hand, causes the fluid velocity to decrease. Furthermore, when comparing the $AA7072$ alloy to the $AA7075$ alloy, fluid velocity is quite slow. Figure 6 depicts the oscillation in $f'(\eta)$ with various values of $\phi$ for both alloys. The increase in $\phi$ lowers the $f'(\eta)$. The velocity $f'(\eta)$ of both alloys decreases as the volume fraction rises. Furthermore, when compared to the $AA7075$ alloy, the velocity of the $AA7072$ alloy is strongly motivated by the volume fraction and falls faster. Figure 7 depicts the behaviour of $f'(\eta)$ in relation to the Weissenberg number $We$. The velocity of the liquid is observed to be reduced across the entire domain as the Weissenberg number rises. Furthermore, when the Weissenberg number increases, the velocity layer thickness decreases. Mathematically, the Weissenberg number $We$ is used in the investigation of viscoelastic flows. It is the ratio of viscous and elastic forces. As a result, as the Weissenberg number rises, the viscous forces diminish, and the velocity profile rises. Figure 8 indicates the effects of $\beta$ on $\theta_1(\eta)$ for both alloys. This indicates that an increase in $\beta$ values significantly improves the temperature profile $\theta_1(\eta)$. This is due to the fact that as the tension between the fluid particles boosts, too much heat is produced, resulting in higher fluid temperatures. Furthermore, for both $AA7072$ and $AA7075$ alloys, the inter-relevance thickness of the thermal layer is increased. Additionally, in $AA7075$ and when treated with $AA7072$ alloy, the closeness of the thermal layer further improves. Figure 9 shows that as $Pr$ increases, so does the temperature of the fluid $\theta_1(\eta)$. According to the observations, the thickness of the boundary layer appears to decrease as $Pr$ increases. As a result, as the Prandtl number upsurges, so does the rate of thermal conductivity. $Pr$ is the ratio of thermal diffusivity and momentum diffusivity. As a result, with a higher $Pr$, heat will dissipate from the sheet more quickly. Fluids with a higher $Pr$ have a lower thermal conduction value. As a result, the $Pr$ attempts to improve the cooling behaviour of the flows. The effect of $\lambda$ on the $\theta_1(\eta)$ profile is portrayed in Figure 10. It shows that as the value of $\lambda$ increases, the temperature field decays. Additionally, for booming values of $\lambda$, the inter-relevance thickness is reduced for both alloys. Furthermore, heat abatement is enhanced in $AA7072$ alloy when treated with $AA7075$ alloy. The fluctuation in the thermal gradient for various values of $\delta_e$ for both alloys is shown in Figure 11. The thermal distribution is enhanced when the values of the thermal relaxation parameters are increased. The heat flow relaxation time causes this parameter to emerge physically. The higher the $\delta_e$ value, the longer it takes for the liquid particles to exchange heat with nearby particles, resulting in a decrease in temperature but an improvement in the temperature gradient. Figure 12 describes the outcome of volume fraction $\phi$ on heat transport in both alloys. The heat transmission of both alloys is improved as the volume fraction increases. Furthermore, in $AA7075$ and when treated with $AA7072$ alloy, the closeness of the thermal layer further improves. Figure 13 depicts the effect of $\sigma$ on $\chi_1(\eta)$ in both alloys. This figure confirms that $\chi_1(\eta)$ has a decreasing nature for various $\sigma$ values, and an increase in the reaction rate parameter $\sigma$ diminishes the concentration of the liquids. In fact, as the reaction rate parameter values increase, the concentration field and related boundary layer thickness decreases. According to Figure 14, a higher Schmidt number corresponds to a lower solute diffusivity, allowing for a shallower penetration of the solute effect. As a result, as $Sc$ rises, $\chi_1(\eta)$ falls. Thus, with lower concentrations of $Sc$, the solute boundary layer is thicker, and vice versa.

Figure 15 depicts the variants in surface drag force $C_{f_x}$ versus $We$ for various $\phi$ values for both alloys. It has been discovered that significantly greater values of $\phi$ enhance the surface drag force, whereas contrasting actions are observed for growing values of $\beta$; see Figure 16. Figure 17 shows the outcome of $\delta_e$ on the rate of heat transfer versus $We$ for both alloys $AA7072$ and $AA7075$. In both $AA7072$ and $AA7075$ alloys, an increase in $\delta_e$ degrades the Nusselt number. Figure 18 illustrates the importance of $\phi$ on $Re_x^{-1/2}Nu_x$ versus $We$ for both $AA7072$ and $AA7075$ alloys. For both alloys, boosting the $\phi$ values improves the heat transmission rate. Figure 19 depicts the variation in $Re_x^{-1/2}Nu_x$ versus $We$ for various $\beta$ values. It can be observed that significantly higher values of $\beta$ enhance the heat transmission rate. Figure 20 depicts the variation in $Re_x^{-1/2}Sh_x$ versus $Sc$ for

various $\phi$ values. It can also be observed that significantly higher values of $\phi$ enhance the concentration rate. A comparison between previous and present works for the validation of the results for skin friction is presented in Table 5.

**Table 5.** Comparison of $-f''(0)$ with literature.

| Published Papers | $-f''(0)$ |
|---|---|
| Zeeshan and Majeed [36] | 0.6058427 |
| B.C Prsannakumara [37] | 0.6069352 |
| Present results | 0.603457 |

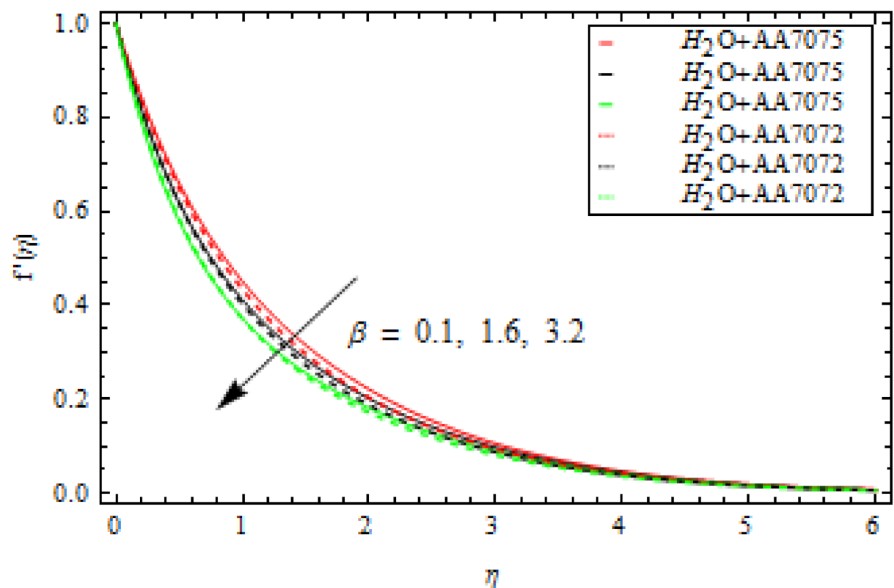

**Figure 5.** Influence of ferromagnetic interaction parameter $\beta$ on velocity profile.

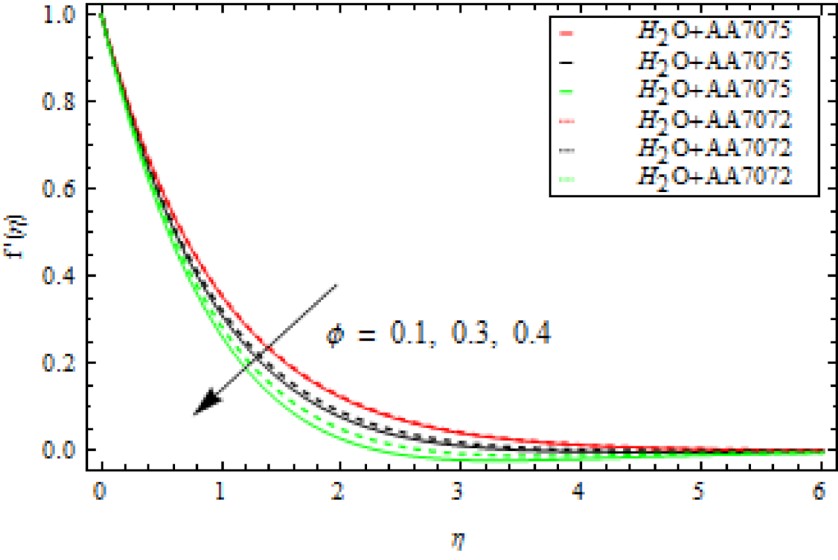

**Figure 6.** Influence of volume fraction $\phi$ on velocity profile.

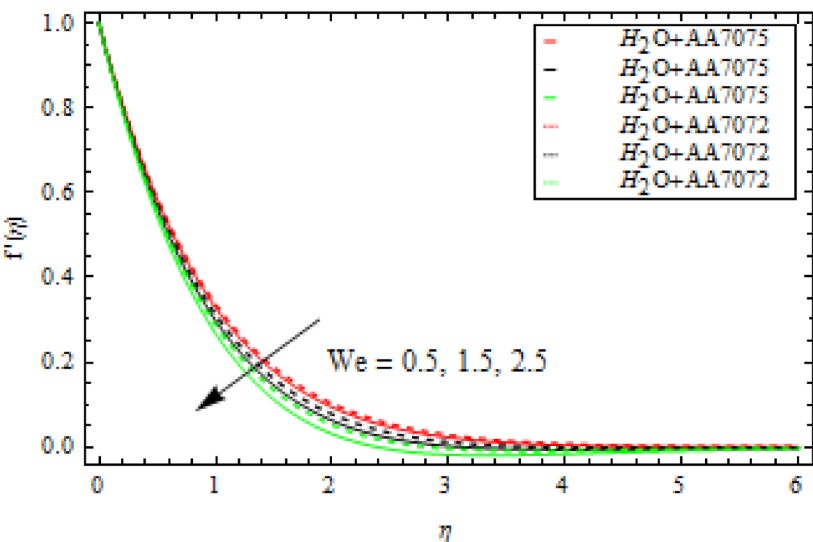

**Figure 7.** Influence of Weissenberg number We on velocity profile.

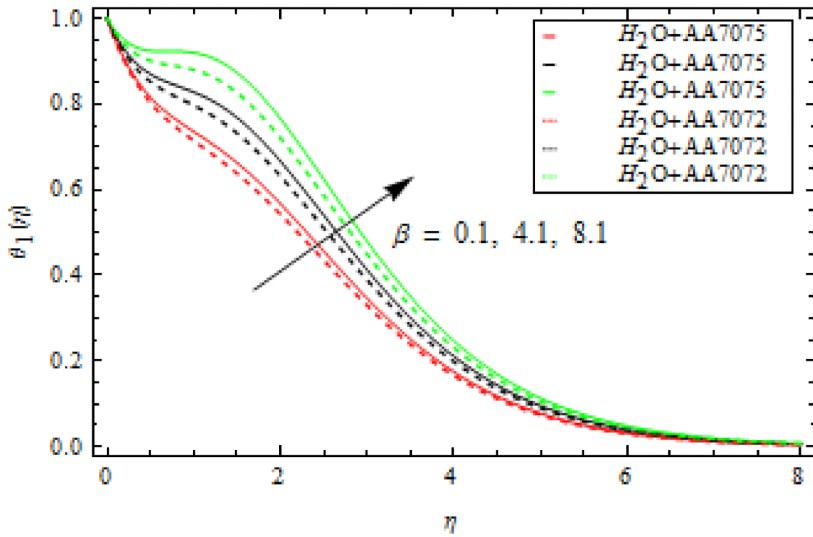

**Figure 8.** Influence of ferromagnetic interaction parameter $\beta$ on temperature profile.

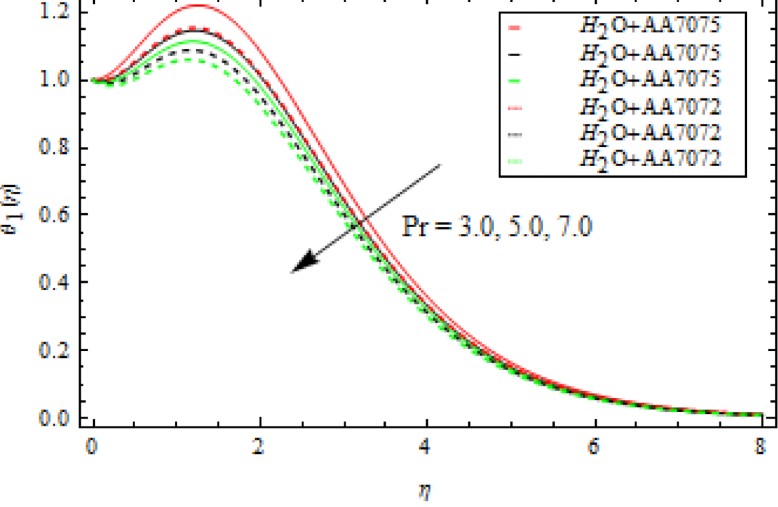

**Figure 9.** Influence of Prandtl number Pr on temperature profile.



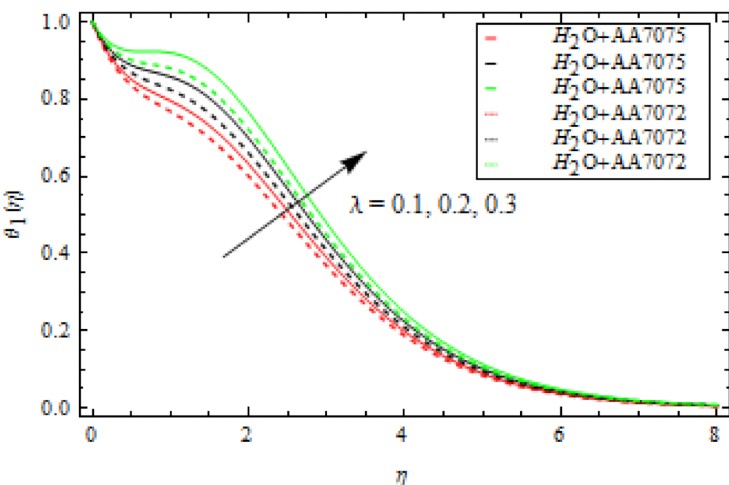

**Figure 10.** Influence of viscous dissipation parameter $\lambda$ on temperature profile.

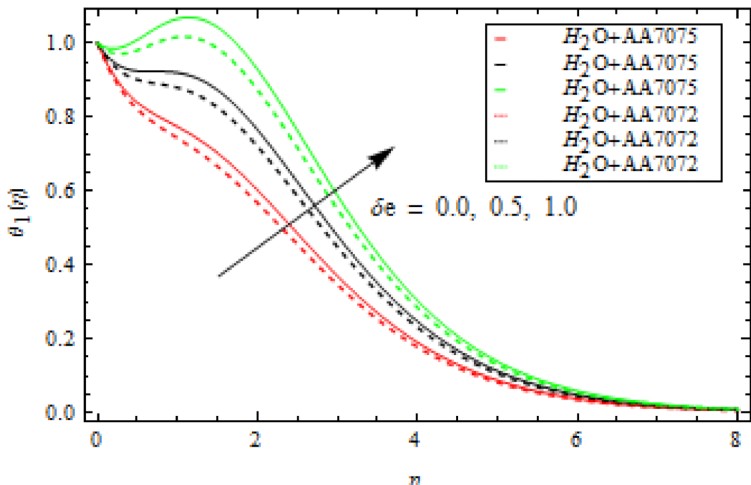

**Figure 11.** Influence of thermal relaxation parameter $\delta_e$ on temperature profile.

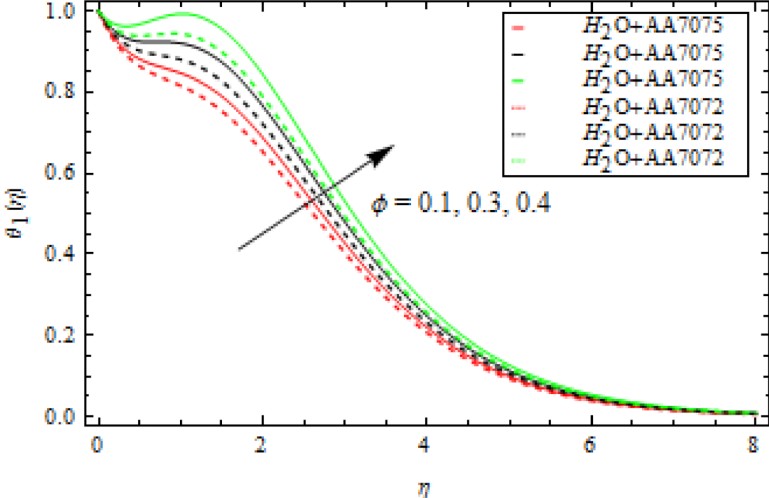

**Figure 12.** Influence of volume fraction $\phi$ on temperature profile.

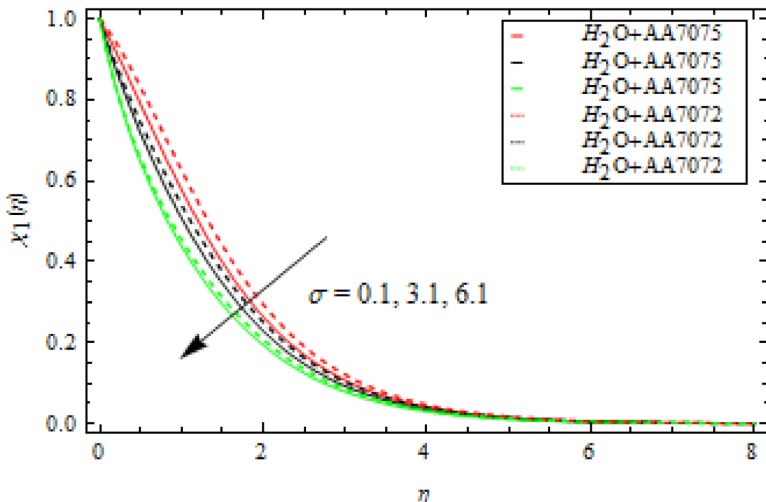

**Figure 13.** Influence of reaction rate parameter $\sigma$ on concentration profile.

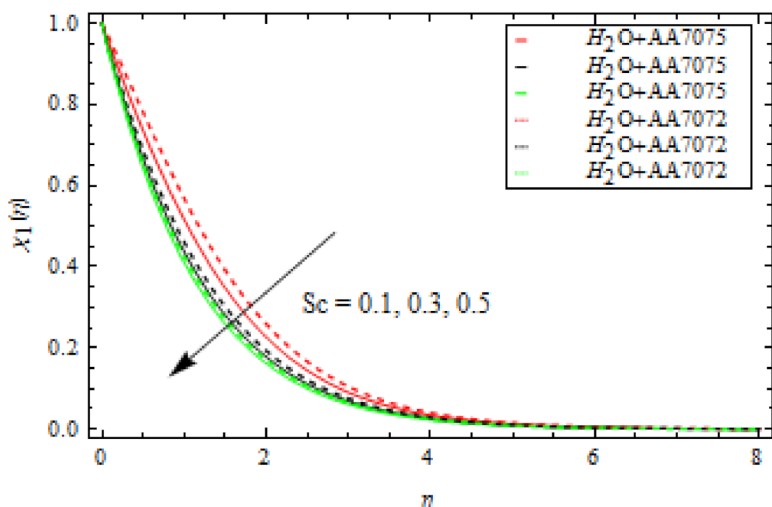

**Figure 14.** Influence of Schmidt number Sc on concentration profile.

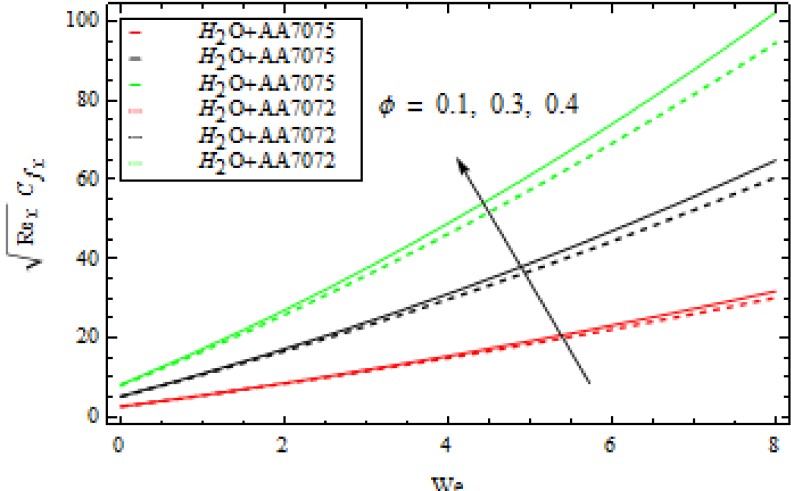

**Figure 15.** Various values of $\phi$ versus We for skin friction.

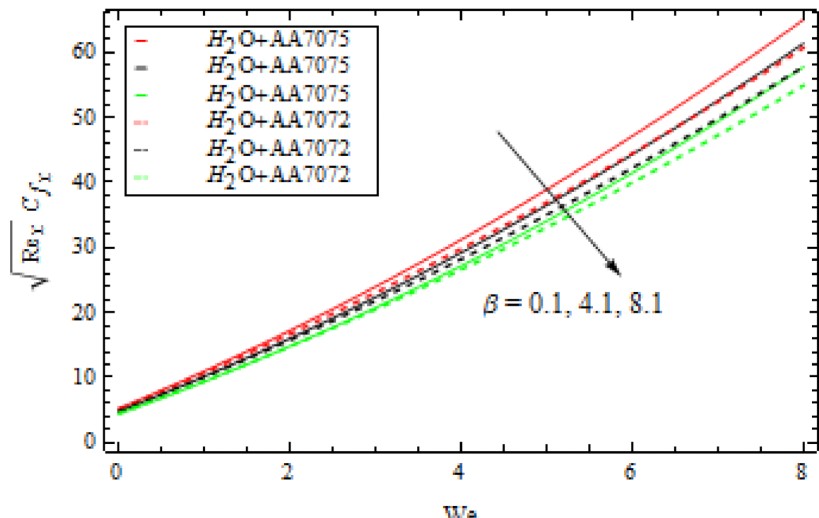

**Figure 16.** Various values of $\beta$ versus We for skin friction.

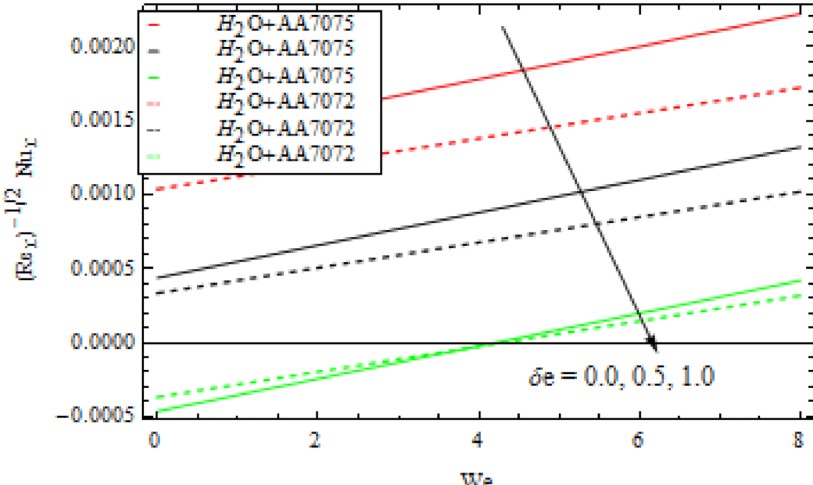

**Figure 17.** Various values of $\delta_e$ versus We.

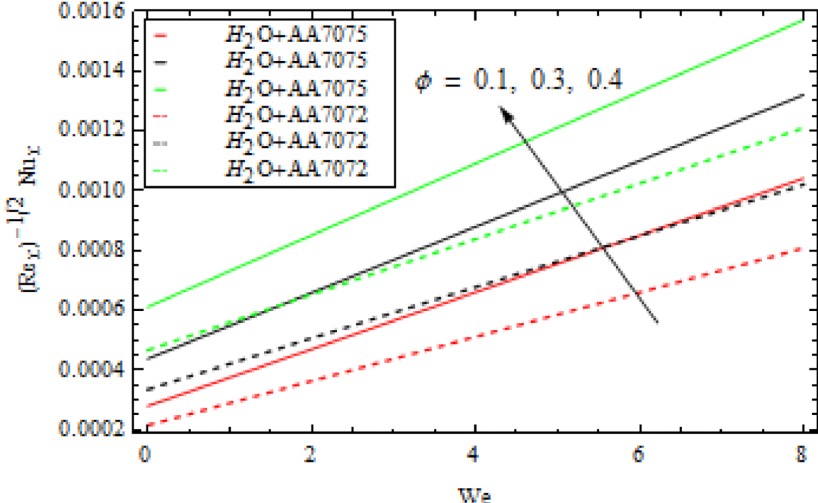

**Figure 18.** Various values of $\phi$ versus We for nusselt number.

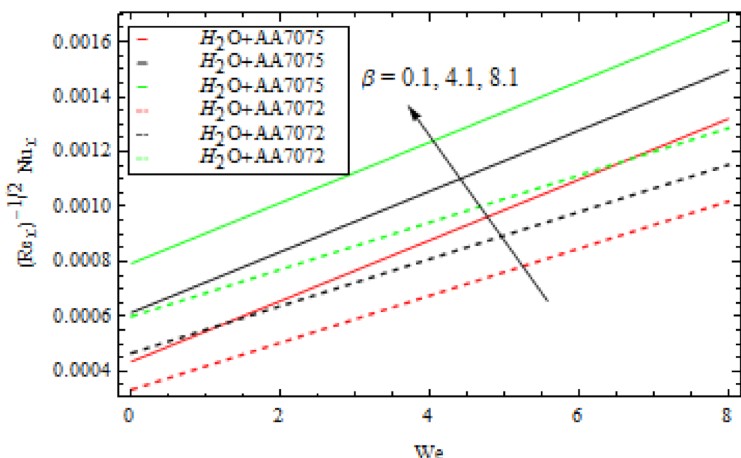

**Figure 19.** Various values of $\beta$ versus We for nusselt number.

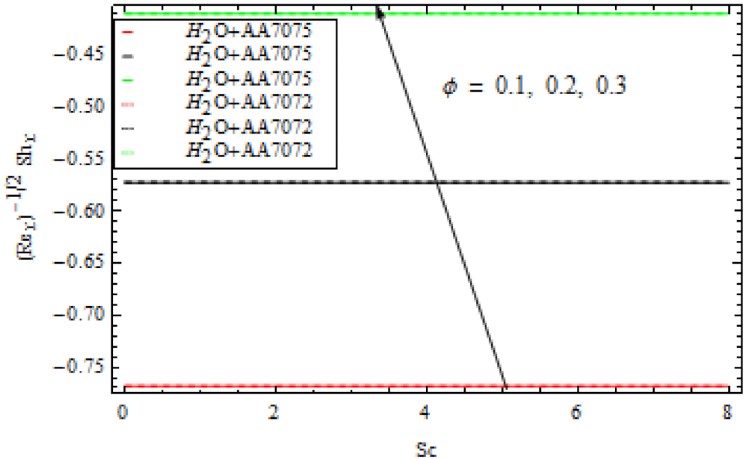

**Figure 20.** Various values of $\phi$ versus Sc.

### 5. Conclusions

Using the influence of magnetic dipoles and the Koo-Kleinstreuer model, the momentum, heat transfer, and mass transfer behavioru of Ree-Eyring nanoliquids through a stretching surface are investigated in this research. Moreover, the heat transmission is described by the Cattaneo-Christov heat flux model, and viscous dissipation is taken into account. Finally, the constructed governing equations related to the momentum, thermal, and mass distributions are converted to ODEs and solved with the HAM. The following are the results of the present analysis:

1.  An escalation in volume fraction, Weissenberg number, and the ferromagnetic interaction parameter affects the velocity gradient. Furthermore, all these parameters negatively influence the velocity gradient of alloy $AA7075$, which falls quicker than the velocity gradient of alloy $AA7072$.

2.  As the ferromagnetic interaction, viscous dissipation parameter, thermal relaxation parameter, and volume fraction grow, the temperature gradient of both alloys increases, whereas contrasting behaviour is revealed for the Prandtl number. Moreover, in $AA7075$ and when treated with $AA7072$ alloy, the closeness of the thermal layer further improves.

3.  A growth in the reaction rate parameter and the Schmidt number brings down the concentration profile. Similarly, all parameters negatively influence the concentration profile of alloy $AA7075$, which drops quicker than the concentration profile of alloy $AA7072$.

4. An improvement in the volume fraction enhances the surface drag force; however, an improvement in the ferromagnetic interaction decreases the surface drag force.
5. The Nusselt number rises as the volume fraction and ferromagnetic interaction grow; however it falls as the thermal relaxation parameter rises.

**Author Contributions:** Conceptualization, Z.S.; Data curation, M.R., W.D. and M.S.; Formal analysis, Z.S., M.R. and W.D.; Funding acquisition, N.V.; Investigation, Z.S.; Methodology, Z.S., M.R. and M.S.; Project administration, Z.S. and M.S.; Resources, Z.S. and N.V.; Software, Z.S., M.R., W.D. and M.S.; Supervision, Z.S.; Validation, Z.S., N.V., W.D. and M.S.; Visualization, Z.S., N.V., M.R. and M.S.; Writing—original draft, Z.S. and W.D.; Writing—review & editing, Z.S. and N.V. All authors have read and agreed to the published version of the manuscript.

**Funding:** The project was financed by Lucian Blaga University of Sibiu and Hasso Plattner Foundation research Grants LBUS-IRG-2021-07.

**Institutional Review Board Statement:** Not applicable.

**Informed Consent Statement:** Not applicable.

**Data Availability Statement:** The data that support the findings of this study are available from the corresponding author upon reasonable request.

**Conflicts of Interest:** The authors declare no conflict of interest.

## Nomenclature

| | |
|---|---|
| $a$ | Distance |
| $c$ | Constant |
| $C_p$ | Specific heat transfer $\left(\text{J·kg}^{-1}\text{·K}^{-1}\right)$ |
| $C$ | Concentration |
| $k$ | Thermal conductivity $\left(\text{W·m}^{-1}\text{·K}^{-1}\right)$ |
| $K$ | Constant |
| $H$ | Magnetic field intensity $\left(\text{A·m}^{-1}\right)$ |
| $M$ | Magnetisation $\left(\text{A·m}^{-1}\right)$ |
| $Pr$ | Prandtl number |
| $Re$ | Local Reynolds number |
| $Sc$ | Schmidt number |
| $T$ | Temperature of fluid |
| $u, v$ | Velocity components $\left(\text{m·s}^{-1}\right)$ |
| $We$ | Weissenberg number |
| $x, y$ | Coordinates axis (m) |
| Greek Letter | |
| $\alpha$ | Dimensionless distance |
| $\beta$ | Ferromagnetic interaction parameter |
| $\gamma$ | Constant |
| $\beta_1, \epsilon$ | Material constant of the fluid |
| $\delta$ | Dimensionless Curie temperature |
| $\delta_e$ | Thermal relaxation parameter |
| $\eta, \xi$ | Independent coordinate |
| $\theta_1(\eta), \theta_2(\eta)$ | Dimensionless temperature profile |
| $\lambda$ | Viscous dissipation parameter |
| $\lambda_2$ | Thermal relaxation time |
| $\mu$ | Dynamic viscosity |
| $\mu_0$ | Magnetic permeability |
| $v$ | Kinematic viscosity |
| $\rho$ | Density $\left(\text{kg·m}^{-3}\right)$ |
| $\rho C_p$ | Heat capacitance |
| $\sigma$ | Reaction rate parameter |

| $\phi_1$ | Scalar potential |
|---|---|
| $\phi$ | Volume fraction |
| $\chi_1(\eta)$, $\chi_2(\eta)$ | Dimensionless concentration profile |
| $\psi$ | Stream function $(\text{m}^2 \cdot \text{s}^{-1})$ |
| Subscript | |
| $f$ | Fluid |
| $nf$ | Nanofluid |
| $c$ | Curie |
| $w$ | Surface |
| $s$ | Solid particle |

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
