# Peer review of "Mathematical Modelling of Ree-Eyring Nanofluid Using Koo-Kleinstreuer and Cattaneo-Christov Models on Chemically Reactive AA7072-AA7075 Alloys over a Magnetic Dipole Stretching Surface"

_coatings, doi:10.3390/coatings12030391_

Round 1

Reviewer 1 Report

The authors investigated the momentum, heat transfer, and mass transfer behavior of Ree-Eyring nanoliquids through a stretching surface using the influence of magnetic dipoles and Koo and Klein-streuer model. I suggest the paper for major revision. The suggested revisions can be found in the following:

  • Please clearly explain in the introduction section that what is the novelty of this article.
  • In section 2, use a graphical figure to clearly show the geometry of the problem used to be solved in this research.
  • In page 3, line 142, it is said that Figure 1 depicts the flow geometry; however, figure 1 is a graph showing the influence of ferromagnetic interaction parameter β on velocity profile. Please modify it.
  • In section 2, for presenting the models (for example: Koo and Kleinstreuer model in page 4) also mention the references.
  • In the nomenclature of the article, please also mention the units of the parameters.
  • In page 4, line 171, what do you mean by this phrase “We succeed in this”? please modify the phrase.
  • Please clearly explain how did you validated your method? This study lacks a validation case.
  • Please explain that how did you discretized the domain.
  • The discussion in section 3 is very shallow. The authors mostly tried to report the graphs. Please discuss and analyze your results in more depth.
  • The quality of figures are very low. Please use better figures with higher resolutions.

Author Response

Responses to Reviewer’s/Editor Comments

Many thanks for the valuable suggestions/comments given by the reviewers/editors to improve the quality of our research manuscript.

Special thanks to you for your painstaking and constructive reviewing work. All the team members speak highly of your responsible attitude in reviewing our paper. Note that if the modifications made according to your comments only, yellow will be used in highlighting this revision.

Responses against each point raised by you are listed as follows

The authors investigated the momentum, heat transfer, and mass transfer behavior of Ree-Eyring nanoliquids through a stretching surface using the influence of magnetic dipoles and Koo and Klein-streuer model. I suggest the paper for major revision. The suggested revisions can be found in the following:

  • Please clearly explain in the introduction section that what the novelty of this article is.

Response: We edit the introduction section and added the novelty of this work please see the last paragraph.

  • In section 2, use a graphical figure to clearly show the geometry of the problem used to be solved in this research.

Response: Thank you for your valuable comments. We added graphical figure see figure.1.

  • In page 3, line 142, it is said that Figure 1 depicts the flow geometry; however, figure 1 is a graph showing the influence of ferromagnetic interaction parameter β on velocity profile. Please modify it.

Response:  Thank you very much for your valuables comments this is a typo mistake we corrected it.

  • In section 2, for presenting the models (for example: Koo and Kleinstreuer model in page 4) also mention the references.

Response: A reference has been added.

  • In the nomenclature of the article, please also mention the units of the parameters.

Response: Nomenclature is edit accordingly, units of the parameters are mentioned

  • In page 4, line 171, what do you mean by this phrase “We succeed in this”? please modify the phrase.

Response: Thank you for your suggestion we modify the phrase.

  • Please clearly explain how did you validated your method? This study lacks a validation case.

Response: Validation of the method is added in the revise manuscript. For solution of the problem we used homotpy analysis method and then compared with ND-solve method. Comparison table and graphs are added please see tables 2-4 and figures 2-4

  • Please explain that how did you discretized the domain.

Response:  As the discretization of the domain allows us calculate approximate solutions over each subdomain rather than over the entire domain.In order to solve Eqs. (14-18) under the boundary conditions (20), we use the Homotopy Analysis Method (HAM) with the procedure given in section 3. The solutions having the auxiliary parameters  adjust and control the convergence of the solutions. The domain here selected from 0 to 6 and for different parameter the domain is selected different. We chose the domain to get best convergence and good results.  

  • The discussion in section 3 is very shallow. The authors mostly tried to report the graphs. Please discuss and analyze your results in more depth.

Response: The discussion section is edit and the results obtained are analyzed properly.

  • The quality of figures is very low. Please use better figures with higher resolutions.

Response: The quality of figures are improved and separate file of all figures are attached, as the journal can set accordingly

Reviewer 2 Report

  1. Should include the physical model to help the readers understand the problem under consideration.
  2. What are the different between this study and the previous studies?
  3. Should make comparison with previously published results.
  4. Should include a section to discuss the numerical method used in this study.
  5. The discussion should include examples of some real applications.
  6. 2 shows the results for volume fraction 10%, 30% and 40% of nanoparticles, which are not physically realistic. Should refer to the experimental results for reasonable percentage of the volume fraction.
  7. The discussion lacks physical interpretation.
  8. 5 shows temperature overshoot near the surface, especially for small values of Pr. What are the reasons behind this phenomenon? Is this case physically realistic?
  9. It is hard to differentiate the curves shown in Fig. 10. Why the concentration overshoot only for the red curves, not the green and black curves?
  10. The concentration equation is given in Eq. (4). So, the quantities of physical interest should also include the local Sherwood number, not only the skin friction coefficient and the local Nusselt number as given in Eqs. (23) and (24).
  11. The results of this study are not complete without the local Sherwood number.
  12. Nomenclature should be arranged in alphabetical order, and the Greek Symbols should be separated. This will help the readers in finding the symbols.

Author Response

Responses to Reviewer’s/Editor Comments

Many thanks for the valuable suggestions/comments given by the reviewers/editors to improve the quality of our research manuscript.

Special thanks to you for your painstaking and constructive reviewing work. All the team members speak highly of your responsible attitude in reviewing our paper. Note that if the modifications made according to your comments only, yellow will be used in highlighting this revision.

Responses against each point raised by you are listed as follows

  1. Should include the physical model to help the readers understand the problem under consideration.

Response: Thank you physical model is included in the revise version

  1. What are the different between this study and the previous studies?

Response: We investigate in this article is that the existence of magnetic dipole and the Koo and Kleinstreuer model are studied using different alloys over a stretching sheet first time. The Cattaneo-Christov model is used to calculate heat transfer in a two-dimensional flow of Ree-Eyring nanofluid across a stretching sheet.the details novelty is highlighted in the revise version

  1. Should make comparison with previously published results.

Response: Comparison with methods and also with previous work is added in the revised manuscript. Please see table 2-5

  1. Should include a section to discuss the numerical method used in this study.

Response: solution method analytical and numerical both are included in details please see section 3 and 3.1.

  1. The discussion should include examples of some real applications.

Response: we edit the discussion section accordingly

  1. 2 shows the results for volume fraction 10%, 30% and 40% of nanoparticles, which are not physically realistic. Should refer to the experimental results for reasonable percentage of the volume fraction.

Response: Thank you for your valuable comments. The present research work is theoretical study not experimental and the volume fraction valued are consider according to the assumption which provide good results.

  1. The discussion lacks physical interpretation.

Response: The discussion section is edit and improved. We added physical interpretation in discussion section.

  1. 5 shows temperature overshoot near the surface, especially for small values of Pr. What are the reasons behind this phenomenon? Is this case physically realistic?

Response: According to the observations, the thickness of the boundary layer appears to decrease as  increases. As a result, as the Prandtl number upsurges, so does the rate of thermal conductivity.  is the ratio of thermal diffusivity and momentum diffusivity. As a result, with higher , heat will dissipate from the sheet more quickly. Fluids with a higher Prandtl number  have a lower thermal conduction value. As a result,  is attempting to improve the cooling behavior of the flows. The details are shown in the revise version

  1. It is hard to differentiate the curves shown in Fig. 10. Why the concentration overshoot only for the red curves, not the green and black curves?

Response:

This figure show the effect of of reaction rate parameter on concentration profile in both alloys cases. This figure confirms that concentration profile has a decreasing nature for various reaction rate parameter values, and an increase in the reaction rate parameter reaction rate parameter diminishes the concentration of the liquids. In fact, as the reaction rate parameter values increase, the concentration field and related boundary layer thickness decreases. the concentration overshoot for the red curve because of its value.

  1. The concentration equation is given in Eq. (4). So, the quantities of physical interest should also include the local Sherwood number, not only the skin friction coefficient and the local Nusselt number as given in Eqs. (23) and (24).

Response: It was typo mistakes. It is corrected and the Sherwood number is added.

  1. The results of this study are not complete without the local Sherwood number.

Response: we did it and it is rectified

  1. Nomenclature should be arranged in alphabetical order, and the Greek Symbols should be separated. This will help the readers in finding the symbols.

Response: Thank you for this suggestion, it is done accordingly.

Reviewer 3 Report

The manuscript ID coatings-1619888 mainly presents numerical and theoretical studies about particular non-Newtonian nanofluids. Heat transfer and magnetic effects exhibited by the samples are analyzed. Please see below a list of comments for the authors:

  1. From the introduction, it is not justified the panoramic research that clearly pointed out the importance of the study of nanoscale aluminium alloys suspended in a fluid.
  2. Several asseverations throughout the text require to be supported by a citation.
  3. It is not clear how were selected the models proposed to describe nanoscale effects since the equations are not provided by proper citations. Contrasting models can be seen in https://doi.org/10.3389/fenrg.2021.767751
  4. In table 1 are shown the parameters related to the base fluid’s and nanoparticles material properties employed in the calculations; however, these parameters were estimated in ref 28 for particular physical conditions that must be fulfilled to calculate other effects. Please argue.
  5. The main results must be confronted in the discussion section with updated publications related to heat transfer and magnetic effects exhibited by comparative samples.
  6. The results plotted in all the results plotted from figures 1 to 15 must be analyzed and explained.
  7. A discussion about the importance in the concentration of the nanoparticles in the nanofluids reported, would improve the presentation of the studied effects. The authors are invited to see the analysis described by https://doi.org/10.1016/j.ijleo.2019.01.042
  8. In my opinion, the initial sentence “Nanoparticles with advanced thermal patterns” should be edited in order to precisely express what the authors want to say.
  9. All the equations must be numbered.
  10. Most of the parameters in the equations reported are not described.

Author Response

Responses to Reviewer’s/Editor Comments

Many thanks for the valuable suggestions/comments given by the reviewers/editors to improve the quality of our research manuscript.

Special thanks to you for your painstaking and constructive reviewing work. All the team members speak highly of your responsible attitude in reviewing our paper. Note that if the modifications made according to your comments only, yellow will be used in highlighting this revision.

Responses against each point raised by you are listed as follows

The manuscript ID coatings-1619888 mainly presents numerical and theoretical studies about particular non-Newtonian nanofluids. Heat transfer and magnetic effects exhibited by the samples are analyzed. Please see below a list of comments for the authors:

  1. From the introduction, it is not justified the panoramic research that clearly pointed out the importance of the study of nanoscale aluminium alloys suspended in a fluid.

Response: We edit the introduction section according to the valuable comments

  1. Several asseverations throughout the text require to be supported by a citation.

Response: we edit and corrected the manuscript accordingly

  1. It is not clear how were selected the models proposed to describe nanoscale effects since the equations are not provided by proper citations. Contrasting models can be seen in https://doi.org/10.3389/fenrg.2021.767751

Response: thank you, it is done accordingly

  1. In table 1 are shown the parameters related to the base fluid’s and nanoparticles material properties employed in the calculations; however, these parameters were estimated in ref 28 for particular physical conditions that must be fulfilled to calculate other effects. Please argue.

Response: Table show the the base fluid’s and nanoparticles material properties and it is not calculated here, it is taken from the literature and references are given.

  1. The main results must be confronted in the discussion section with updated publications related to heat transfer and magnetic effects exhibited by comparative samples.

Response: The main results are updated accordingly

  1. The results plotted in all the results plotted from figures 1 to 15 must be analyzed and explained.

Response: all figures are analyzed and explain properly and more results are added

  1. A discussion about the importance in the concentration of the nanoparticles in the nanofluids reported, would improve the presentation of the studied effects. The authors are invited to see the analysis described by https://doi.org/10.1016/j.ijleo.2019.01.042

Response: Thank you it is done accordingly.

  1. In my opinion, the initial sentence “Nanoparticles with advanced thermal patterns” should be edited in order to precisely express what the authors want to say.

Response:  it is corrected, thanks

  1. All the equations must be numbered.

Response: rectified

  1. Most of the parameters in the equations reported are not described.

Response: we edit the manuscript and added the parameter in nomenclature.

Round 2

Reviewer 1 Report

Dear Editor,
The paper is properly modified compared to its previous version. I have no further comments and I suggest the paper for publication. 

Reviewer 2 Report

The authors have revised the manuscript as suggested.

Reviewer 3 Report

The authors have importantly improved the presentation of their work after the review stage. The main results and the analysis can be useful for particular nanofluids, and the manuscript is reported with clear descriptions. Then, in my opinion, this work can be considered for publication as it is.